# HieraQuery: Bridging Multimodal Understanding and High-Quality Generation through Multi-Scale Query Learning

## Abstract

Unified multi-modal LLMs enable the integration of visual understanding and generation in a single framework. Recent study shows that a set of learnable queries can serve as an effective interface between autoregressive multimodal LLMs and diffusion models, though the visual quality of generated images still lag behind dedicated generation models. Actually, it is hard for a single set of learnable queries to generate accurate visual representations in a single round of inference. Hence, we introduce `HieraQuery`, which leverages a hierarchy of learnable visual queries to generate high-quality visual contents in a coarse-to-fine manner. Specifically, several sets of learnable queries are provided to the language model, where preceding ones are used to generate images of lower resolution, focusing on the global structures of the generated content, while the subsequent ones serve as the condition for generating higher resolution images, concentrating on the fine-grained details. In addition, a multi-scale representation alignment strategy is proposed to enforce cross-scale consistency and accelerate convergence. Ablation analyses demonstrate that using the hierarchical visual queries can effectively improve the visual generation capability of unified multi-modal LLMs, and scaling up the number of scales proves an effective way for further improving the generation quality.

## 1 Introduction

The release of GPT-4o (OpenAI, 2025) in March 2025, which introduced native image generation, leads to prompt attention to unified models for understanding and generation (Tong et al., 2024; Team, 2024b; Pan et al., 2025). Users can now perform complex visual tasks such as image editing (Gong et al., 2024a; Feng et al., 2024), multi-view synthesis (Shi et al., 2023), style transfer (Li et al., 2024b), and even 3D rendering (Mildenhall et al., 2021) purely through natural language conversations. These capabilities are readily accomplished by specialized models (Gong et al., 2024b; Wang et al., 2024a; Huang et al., 2024a; Tan et al., 2024a), marking a major advance in vision-language intelligence. A major challenge in unifying multimodal understanding and generation lies in the inconsistency of visual feature spaces. Recent models such as TokenFlow (Qu et al., 2024) and Janus (Wu et al., 2024a) integrate diffusion-based decoders with token-based understanding models, achieving strong image generation but often at the cost of less precise understanding.

One crucial problem solved recently for image generation in unified autoregressive models is error accumulation. To tackle this issue, a set of learnable queries is employed to generate all continuous conditions simultaneously (Zhang et al., 2025; Pan et al., 2025). However, achieving high-quality generation and editing of visual content demands accurate generation of these continuous condition tokens, which remains difficult for models that produce all tokens at a single inference step.

Inspired by Chain-of-Sight (Huang et al., 2024b) and VAR (Tian et al., 2024), which exploit multi-scale visual structures to capture details at varying spatial scales for understanding and generation tasks, we propose `HieraQuery`, a unified autoregressive model capable of both visual understanding and generation at high precision. We present two key designs to effectively leverage multi-scale hierarchy for visual generation. The first is the multi-scale query learning. Several sets of learnable query tokens are sequentially connected and fed to the language model. The corresponding outputs

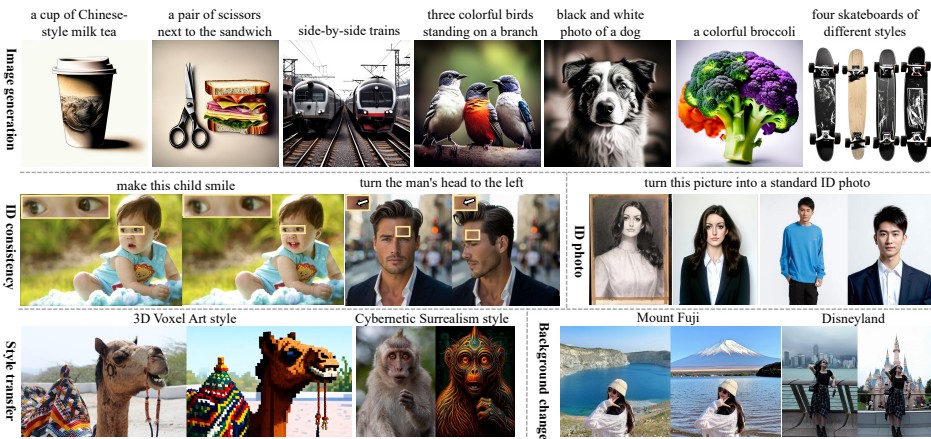

Figure 1: **The output results and multimodal interactive demos of `HieraQuery`**. Our model supports text-to-image generation, image editing, and image style transfer following textual instructions.

act as the condition for generating images at different resolutions. The query tokens are hierarchically structured from coarse to fine, with preceding tokens guiding the generation of lower-resolution images and the subsequent tokens refining details for higher-resolution outputs. The second component is the multi-scale representation alignment, where we align the intermediate representations of the diffusion transformer to the latent space of the same visual encoder to encourage cross-scale consistency and accelerate training.

Building on top of the conceptual idea of MetaQuery (Pan et al., 2025), we find `HieraQuery` resolves the previous limitation where an increase in the number of tokens did not notably enhance the visual quality in generation. A consistent improvement in the generation quality is observed when we gradually increase the number of queries with increased number of scales. Ablation analyses shows that the incoporation of `HieraQuery` demonstrates a 10pt improvement in the overall GenEval score when compared to the single scale baseline, while maintaining state-of-the-art understanding capabilites of the base MLLM. Additionally, it is demonstrated that the multi-scale representation alignment strategy notably improves the consistency between the generated images between different scales. Finally, we show `HieraQuery` is also capable of image editing with strong control fluency and contextual understanding. Some of the results are shown in Fig. 1.

## 2 RELATED WORK

**Multimodal Understanding Models**     Large language models (LLMs) leveraging pre-trained transformer architectures (Vaswani et al., 2017) have profoundly transformed natural language processing. The multi-modal systems for visual recognition are further reshaped by integrating with LLMs, where three core architectural components are typically identified: (1) a visual encoder for modality-specific feature extraction (Radford et al., 2021; Zhai et al., 2023; Dehghani et al., 2024), (2) a pre-trained LLM backbone for language understanding (Dubey et al., 2024; Bai et al., 2023; Brown et al., 2020), and (3) adaptive connector modules facilitating cross-modal alignment. Current implementations primarily diverge in the connector designs – early approaches like BLIP employ learnable query tokens for cross-modal interaction (Dai et al., 2023), while later frameworks such as LLaVA (Liu et al., 2024) adopt streamlined linear projection layers. Beyond static image comprehension, contemporary multimodal systems incorporate dynamic visual (video) and auditory modalities through unified sequence modeling approaches, evolving into omni-MLLM architectures (Bai et al., 2025; Guo et al., 2025) capable of processing heterogeneous input combinations.

**Unified Multimodal Large Language Models**     Both visual recognition and generation have benefited from adopting LLMs, thus it is very appealing to explore how to accomplish cross-modal understanding and generation in a unified framework. Pioneering works like Chameleon (Team, 2024a), Show-o (Xie et al., 2024b) and Emu3 (Wang et al., 2024b) have tried to directly adopt the unified VQ tokenizer to encode images for both multimodal understanding and generation. Since

VQ tokenizer usually cannot maintain high understanding performance of MLLMs due to the information loss during quantization, the Janus series (Wu et al., 2024b; Chen et al., 2025b) use separate encoders for understanding and generation. On the other hand, VILA-U (Wu et al., 2024c) and MUSE-VL (Xie et al., 2024c) try to align VQ-tokenizer with continuous VLM features, thereby improving the understanding performance of the model.

Another way to empower language models with image generation ability is to integrate MLLMs with diffusion models. For example, Emu (Sun et al., 2023) uses the LLM output as a condition for the pre-trained diffusion model, Transfusion (Zhou et al., 2024) train a single transformer by combining the cross-entropy loss with diffusion. Recent works further use learnable query tokens to extract semantic information from MLLM and use them as conditions for the diffusion model (Pan et al., 2025; Zhang et al., 2025). This approach enables the framework to have the ability to generate, edit, and communicate while maintaining the understanding performance of MLLM. However, such a framework requires the model to generate highly precise image representations using a single set of learnable queries, which is quite challenging. This proves to be the main bottleneck in further scaling up the generation quality (Pan et al., 2025), as the generation performance rapidly plateaus with the increasing number of learnable queries. Hence, in this work, we aim to make the generation of precise image representations easier, by introducing a hierarchy of visual query tokens that generates the image in a coarse-to-fine manner, thus bringing even more vitality to the community developing unified multi-modal LLMs.

# 3 APPROACH

`HieraQuery` compresses image representations into a sequence of continuous tokens, which are combined with discrete text tokens and further processed by a scaled auto-regressive Transformer (Bai et al., 2025; Lu et al., 2024) for end-to-end multimodal context learning. The general framework of `HieraQuery` follows MetaQuery (Pan et al., 2025), where the generation capability is provided by an external trainable diffusion model, conditioned on tokens produced by auto-regressive Transformers.

To address the difficulty of generating accurate visual representations with a single set of learnable queries in a single inference step, we present two key modifications in our `HieraQuery`. The first is the multi-scale query learning (Sec. 3.1), which enables the generation of visual content from a coarse global structures to fine-grained details. The second is the multi-scale representation alignment strategy (Sec. 3.2) for maintaining the cross-scale consistency between generated images of different resolutions. The overall framework of `HieraQuery` is presented in Fig. 2.

## 3.1 MULTI-SCALE QUERY LEARNING

To overcome the difficulty of directly generating accurate visual conditions for the image at the target resolution, we present multi-scale learnable query tokens in replacement of the learnable queries in (Pan et al., 2025). The multi-scale query tokens are structured from a coarse-to-fine order, where preceding queries are used as conditions for generating low-resolution images and the subsequent ones are used for generating high-resolution images.

**Multi-Scale Learnable Tokens Construction** Given an input image $x$, we define a set of scales $\mathcal{S} = \{s_1, s_2, \ldots, s_K\}$, where each $s_k$ corresponds to a spatial resolution, e.g., $s_k \in \{4 \times 4, 8 \times 8, 16 \times 16\}$. Each scale $s_k$ is associated with a dedicated set of learnable query tokens $Q_{s_k} \in \mathbb{R}^{N_{s_k} \times d}$, where $N_{s_k}$ is the number of tokens for scale $s_k$, and $d$ is the hidden dimension size. Formally, we initialize the multi-scale query tokens as:

$$Q = \{Q_{s_1}, Q_{s_2}, \ldots, Q_{s_K}\}, \quad Q_{s_k} = \text{Learnable Parameters.} \tag{1}$$

Each $Q_{s_k}$ is designed to produce image contents at different granularities. The preceding ones focuses more on the global layout, color distributions and major objects, while the subsequent ones encodes fine-grained textures and detailed patterns. Given that modeling image content becomes progressively more challenging at higher granularities - it's generally easier to model lower resolution images compared to higher resolution ones – the quantity of learnable queries we consider rises as the resolution itself increases.

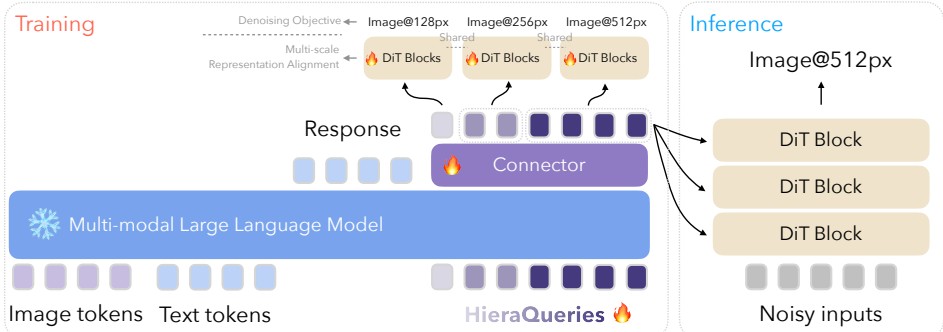

Figure 2: **The framework of `HieraQuery`.** Our model leverages multi-scale learnable query tokens for generating images at different resolutions, greatly reducing the difficulty at modeling high resolution images directly in a single inference step. Further, multi-scale representation alignment is introduced for cross-scale consistency between images of different resolutions.

**Multi-Scale Learnable Tokens Fusion and Processing**    To preserve scale-specific semantics, we introduce explicit scale boundary markers. For each scale $s_k$, we prepend and append special tokens:

$$\text{Input}_{s_k} = [\text{START}_{s_k}, Q_{s_k}, \text{END}_{s_k}], \tag{2}$$

where $\text{START}_{s_k}$ and $\text{END}_{s_k}$ are learnable tokens indicating the boundaries of $s_k$. Each query token also receives a dedicated positional encoding. Let $\mathbf{P}_{s_k} \in \mathbb{R}^{N_{s_k} \times d}$ denote the positional grid encoding for scale $s_k$, constructed according to the spatial grid associated with that resolution. The complete multi-scale token sequence fed into the language model is thus:

$$\mathbf{Z}_{\text{input}} = \text{Concat}\left(\text{Input}_{s_1}, \text{Input}_{s_2}, \ldots, \text{Input}_{s_K}\right) + \text{Concat}\left(\mathbf{P}_{s_1}, \mathbf{P}_{s_2}, \ldots, \mathbf{P}_{s_K}\right). \tag{3}$$

The language model $f_\theta(\cdot)$ processes $\mathbf{Z}_{\text{input}}$ to produce the hidden representations $\mathbf{H}$. The hidden representations are then provided to a trainable connector layer $g_\theta(\cdot)$ for fine-tuning the visual representations before feeding the output representations as the condition to diffusion transformers, which produces images of different resolutions $\hat{\mathbf{X}}_{\mathbf{s_1}}, \hat{\mathbf{X}}_{\mathbf{s_2}}, \hat{\mathbf{X}}_{\mathbf{s_3}}$:

$$\mathbf{H} = f_\theta(\mathbf{Z}_{\text{input}}), \tag{4}$$

$$\hat{\mathbf{X}}_{\mathbf{s_1}}, \hat{\mathbf{X}}_{\mathbf{s_2}}, \hat{\mathbf{X}}_{\mathbf{s_3}} = \text{DiT}(g_\theta(\mathbf{H})). \tag{5}$$

### 3.2  MULTI-SCALE REPRESENTATION ALIGNMENT

To maintain the consistency between generate images of different resolutions, we introduce a simple yet effective Multi-Scale Representation Alignment strategy. This strategy leverages the intrinsic capabilities of unified multi-modal models, which include a visual encoder transforming image pixels into semantically rich representations. Specifically, our approach mirrors (Yu et al., 2024a), by aligning the intermediate hidden states extracted from the DiT backbone at various resolutions with the corresponding outputs of the visual encoder within the MLLM framework. However, due to the potential mismatches between the input image resolution and the DiT backbone resolution, we employ interpolation at the feature level to adjust the shape of the semantic representations to match that of the DiT hidden states prior to the alignment process.

## 4  EXPERIMENTAL SETUP

### 4.1  TRAINING DATA

**Basic Image-Text Pairs**    We use part of LAION-5B (Schuhmann et al., 2022) (372K samples), Wukong (Gu et al., 2022), Midjourney, Blip3o (Chen et al., 2025a), and other datasets that commonly used for diffusion model training. We apply aspect ratio ($\leq 2.5$), watermark detection ($\leq 0.5$), and CLIP alignment ($\geq 0.45$) thresholds for filtering. After preprocessing, the final sample counts are 4M (Wukong), 5M (Midjourney), 3M(Blip3o), and 2M (others).

**Image Generation Datasets** This part of training data includes InstructPix2Pix-clip-filtered (Brooks et al., 2023), where each pair of edited images is generated 100 times, and the best examples are chosem based on CLIP metrics (Sec.3.1.2 in InstructPix2Pix), SEED-Data-Edit-part2/3 (Ge et al., 2024), excluding part1 due to the poor visual quality, Ultra-edit (Zhao et al., 2024), SynCD (Kumari et al., 2025), Subjects200k (Tan et al., 2024b), HQ-edit (Hui et al., 2024), and MagicBrush (Zhang et al., 2023). It consists of 5,008,795 samples. In addition, our training data includes publicly available datasets commonly used for style transfer tasks, along with synthesized data generated using style prompts. The style data comprises a high-quality subset of WikiArt, covering 27 painting styles such as Impressionism, Realism, and Expressionism, and the StyleBooth, featuring 67 styles including cartoon and 3D, each with 717 image pairs. These two datasets contain 81,444 and 80,922 samples, respectively. We selected a total of 2 million relatively high-quality samples from these data sets for training

## 4.2 IMPLEMENTATION DETAILS

**Modules** Based on the previous experience of MetaQuery (Pan et al., 2025), we continue to build generation capabilities on the powerful MLLM. Our experiments are based on Ming-Lite-Omni (InclusionAI, 2025), which is an open-source MoE-based Multimodal LLM, processing audio, video, image, and text inputs to generate multimodal outputs via balanced training strategies, maintaining robust text performance to advance omni-MLLM development. Following (Pan et al., 2025), we adopt Qwen2.5-0.5B (Qwen et al., 2025) as our connector, and initialize the trainable diffusion model with SD3-Medium (Esser et al., 2024) / SANA-1.6B (Xie et al., 2024a). Unless otherwise specified, the diffusion model is initialized using SD3-Medium. In order to make a fair comparison with the existing papers, we also re-trained with the SANA-1.6B initialization. Wherever additional transformations of the hidden-states dimension are required, we use a two-layer multilayer perceptron.

**Training** HieraQuery training process includes three stages: text-to-image pretraining, image reconstruction training, and image editing fine-tuning. Image reconstruction training is a preparation for editing training. The purpose is to make the output of the model consistent with the input, and use the reconstruction dataset converted from image-text pairs for training. Each stage is performed on the training dataset for two epochs, with the cosine learning rate strategy and warmup, and the learning rate increases from 1e-6 to 5e-5 and then decreases to 1e-6. The parameters of MLLM are frozen during the whole process, and all other parameters are trainable, and there is no special learning rate setting.

## 5 EXPERIMENTS

### 5.1 TEXT-TO-IMAGE GENERATION

We conduct separate quantitative evaluations of `HieraQuery` on multimodal understanding and generation using public benchmarks. For multimodal generation, we evaluate text-to-image performance by FID score (Heusel et al., 2017) on MJHQ-30K (Li et al., 2024a) for visual aesthetic quality, and GenEval (Ghosh et al., 2024) and DPG-Bench (Hu et al., 2024) (both without prompt rewriting) for prompt alignment, respectively.

As shown in Tab. 1, our `HieraQuery` obtains a state-of-the-art 0.87 overall accuracy on GenEval without using any LLM rewriter. It can be seen that with a similar level of understanding performance, `HieraQuery` outperforms all other unified models in image generation on MJHQ-FID, GenEval and DPG-bench. Quantitative comparison is shown in Figure 3.

### 5.2 UNDERSTANDING

We compare our model with existing approaches on understanding benchmarsk in Tab. 1, including MMB (Liu et al., 2025b), MMMU (Yue et al., 2024), AI2D (Kembhavi et al., 2016), and MM-Vet (Yu et al., 2024b). Our model demonstrates competitive performance among models of similar size, highlighting its robust capabilities in image-text understanding tasks. Furthermore, our model exhibits competitive performance among models of similar size, showcasing its robust capabilities in image-text understanding tasks.

Table 1: Quantitative results on generation benchmarks and parts of OpenCompass (Contributors, 2023) multimodal leaderboard. ‡ denotes closed-source models. "Und." and "Gen." denote "understanding" and "generation", respectively. † refers to the methods using LLM rewriter.

| Type | Model | MMB ↑ | MMMU↑ | AI2D ↑ | MM-Vet ↑ | MJHQ FID ↓ | GenEval ↑ | DPG-Bench ↑ |
|------|-------|-------|-------|-------|-------|-------|-------|-------|
| *Und.* | LLaVA-72B (Xie et al., 2024b) | 84.5 | 56.6 | 86.2 | 60.6 | × | × | × |
| | Qwen2-VL-72B (Bai et al., 2023) | 85.9 | 64.3 | 88.3 | 73.9 | × | × | × |
| | Qwen2.5-VL-7B (Bai et al., 2025) | 87.8 | 67.9 | 88.2 | 76.7 | × | × | × |
| | Emu3-Chat (Wang et al., 2024c) | 58.5 | 31.6 | - | 37.2 | × | × | × |
| | InternVL2.5-8B (Chen et al., 2024) | 82 | 54.8 | 84.5 | 68.1 | × | × | × |
| | DeepSeek-VL2 (Wu et al., 2024d) | 81.2 | 50.7 | 84.5 | 60.0 | × | × | × |
| | GPT-4o-20241120‡ (OpenAI, 2024) | 84.3 | 70.7 | 84.9 | 74.5 | × | × | × |
| | Step-1o‡ (StepFun, 2025) | 87.3 | 69.9 | 89.1 | 82.8 | × | × | × |
| *Gen.* | LlamaGen (Sun et al., 2024) | × | × | × | × | - | 0.32 | - |
| | LDM (Rombach et al., 2022) | × | × | × | × | - | 0.37 | - |
| | SDv1.5 (Rombach et al., 2022) | × | × | × | × | - | 0.43 | - |
| | PixArt-$\alpha$ (Chen et al., 2023) | × | × | × | × | 6.14 | 0.48 | 71.6 |
| | SDv2.1 (Rombach et al., 2022) | × | × | × | × | - | 0.50 | - |
| | DALL-E 2 (Ramesh et al., 2022) | × | × | × | × | - | 0.52 | - |
| | Emu3-Gen (Wang et al., 2024c) | × | × | × | × | - | 0.54 | - |
| | SDXL (Podell et al., 2023) | × | × | × | × | 6.63 | 0.55 | 74.7 |
| | DALL-E 3 (Betker et al., 2023) | × | × | × | × | - | 0.67 | 83.5 |
| | SD3-Medium (Esser et al., 2024) | × | × | × | × | 11.92 | 0.74 | 84.1 |
| *Unified* | DreamLLM (Dong et al., 2023) | - | - | - | 36.6 | - | - | - |
| | MetaMorph (Tong et al., 2024) | 75.2 | - | - | - | - | - | - |
| | Show-o (Xie et al., 2024b) | - | 26.7 | - | - | 15.18 | 0.53 | - |
| | TokenFlow-XL (Qu et al., 2024) | 68.9 | 38.7 | - | 40.7 | - | 0.55 | 73.3 |
| | Chameleon (Team, 2024b) | - | 22.4 | - | 8.3 | - | 0.39 | - |
| | Janus (Wu et al., 2024a) | 69.4 | 30.5 | - | 34.3 | 10.10 | 0.61 | - |
| | JanusFlow (Ma et al., 2024) | 74.9 | 29.3 | - | 30.9 | 9.51 | 0.63 | 80.09 |
| | Janus-Pro-1B (Chen et al., 2025b) | 75.5 | 36.3 | - | 39.8 | 14.33 | 0.73 | 82.6 |
| | Metaquery-XL (Pan et al., 2025) | 83.5 | 58.6 | - | 66.6 | 6.02 | 0.80† | 82.0 |
| | OmniGen2 (Wu et al., 2025) | 79.1 | 53.1 | - | 61.8 | - | 0.80 | 83.6 |
| | Blip3-o (Chen et al., 2025a) | 83.5 | 58.6 | - | 66.6 | - | 0.84 | 81.6 |
| | BAGEL (Deng et al., 2025) | 85.0 | 55.3 | - | 67.1 | - | 0.82 | - |
| | UniWorld-V1 (Lin et al., 2025) | 83.5 | 58.6 | - | 67.1 | - | 0.80 | - |
| | **Ours (HieraQuery)** | 80.7 | 54.3 | 84.9 | 74.0 | **4.85** | **0.87** | **84.2** |

Table 2: Quantitative evaluation on GEdit-Bench-EN. All metrics are reported as higher-is-better (↑). We report the results evaluated by GPT-4.1. The Intersection subset reflects the subset of prompts where all methods return valid responses with a total of 434 instances; the Full set includes all the 606 instances.

| Model | GEdit-Bench-EN (Intersection subset) ↑ | | | GEdit-Bench-EN (Full set) ↑ | | |
|-------|-------|-------|-------|-------|-------|-------|
| | G_SC | G_PQ | G_O | G_SC | G_PQ | G_O |
| Instruct-Pix2Pix (Brooks et al., 2023) | 3.473 | 5.601 | 3.631 | 3.575 | 5.491 | 3.684 |
| MagicBrush (Zhang et al., 2023) | 4.646 | 5.800 | 4.578 | 4.677 | 5.656 | 4.518 |
| AnyEdit (Yu et al., 2025) | 3.177 | 5.856 | 3.231 | 3.178 | 5.820 | 3.212 |
| OmniGen (Xiao et al., 2025) | 6.070 | 5.885 | 5.162 | 5.963 | 5.888 | 5.061 |
| Step1X-Edit (Liu et al., 2025a) | 7.183 | 6.818 | 6.813 | 7.091 | 6.763 | **6.701** |
| **Ours(HieraQuery)** | **7.633** | **7.097** | **6.849** | **7.357** | **7.102** | 6.616 |
| Gemini (Gemini2, 2025) | 6.697 | 6.638 | 6.322 | 6.732 | 6.606 | 6.315 |
| GPT-4o (OpenAI, 2025) | 7.844 | 7.592 | 7.517 | 7.850 | 7.620 | 7.534 |

## 5.3 INSTRUCTION BASED IMAGE EDITING

As shown in Tab. 2, we use GEdit-Bench to quantitatively test the model's response to various editing instructions. As shown in Tab. 2, HieraQuery excels a wider range of interactive image editing tasks, including style transfer and object addition, deletion, and modification, outperforms existing open-source models in both SQ (Semantic Consistency), PQ (Perceptual Quality), and O (Overall Score). When editing human figures, it demonstrates a clear advantage in maintaining scene and character ID (As shown in Fig. 4).

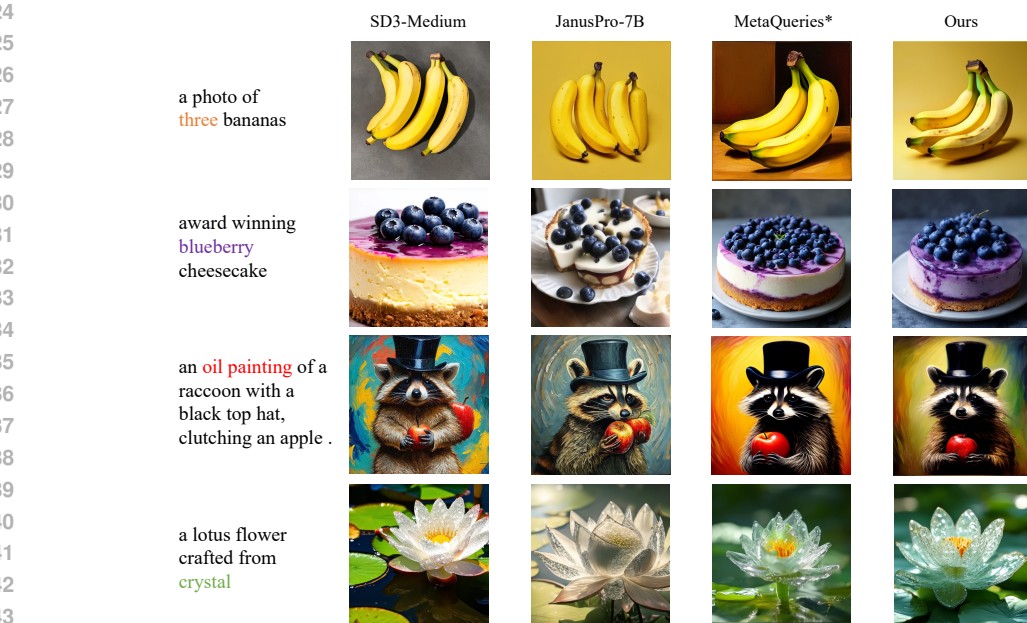

Figure 3: Quantitative comparison on text-to-image generation. Our method excels at prompt following, detail depiction, style accuracy, and subject integrity. * denotes models that are currently not publicly available and are therefore reproduced by us.

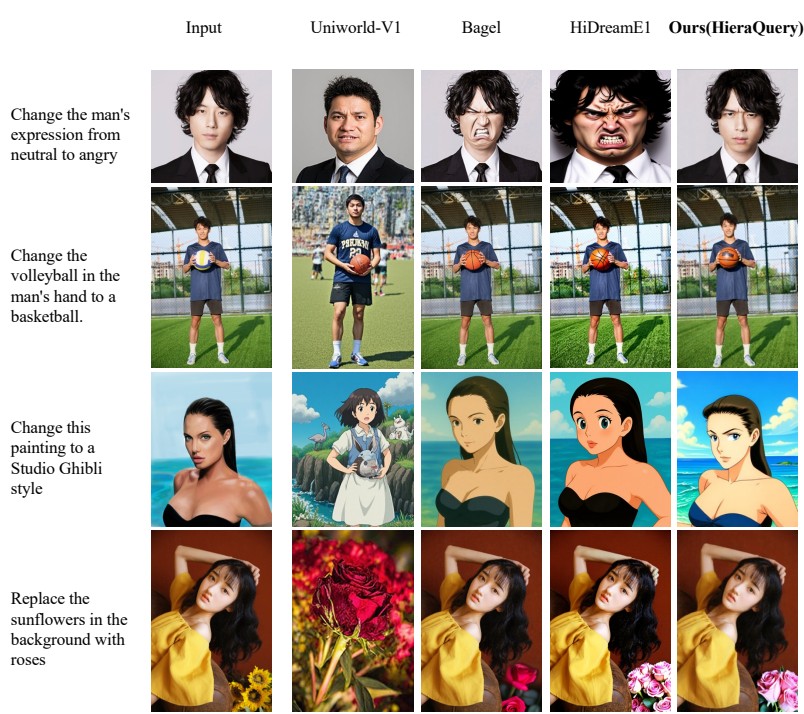

Figure 4: Illustration of image editing results compared to recent approaches such as Uniworld-V1 (Lin et al., 2025), Bagel (Deng et al., 2025), and HiDreamE1 (Cai et al., 2025).

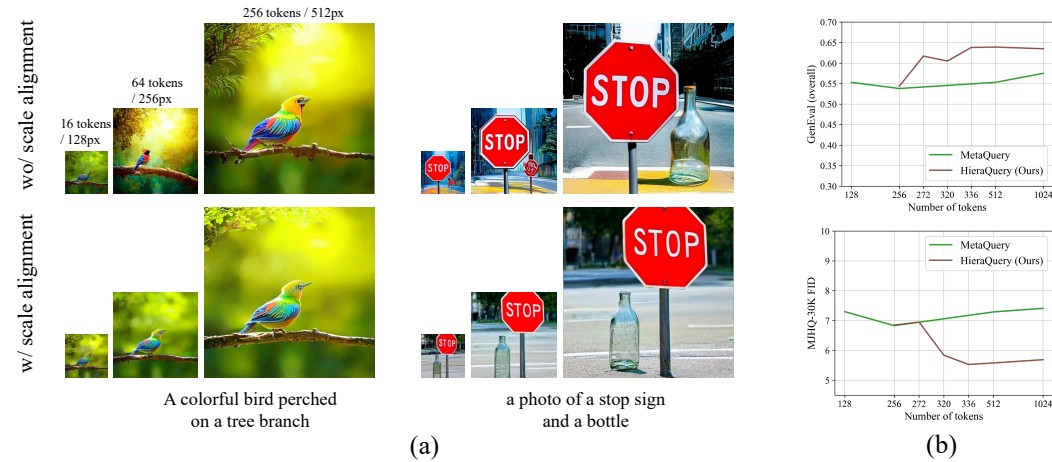

Figure 5: (a) Qualitative comparison of the ablation of multi-scale representation alignment method. The proposed representation alignment method can help tokens of different scales represent similar images, thereby coarse-to-fine improving the final high-resolution visual effect. (b) Study on the scaling of token numbers. As the number of tokens increases, our multi-scale learnable queries setting increases faster than single-scale in image generation performance.

Table 3: Ablation studies on the proposed mothods. "M" denotes *Multi-scale Query Learning*, "A" denotes *Multi-Scale Representation Alignment*. We report all aspects in GenEval (%).

| M. | A. | Num tokens | Single Obj.↑ | Two Obj.↑ | Counting↑ | Colors↑ | Position↑ | Color Attri.↑ | Overall↑ |
|---|---|---|---|---|---|---|---|---|---|
| × | × | 256 | 96.8 | 55.8 | 40.6 | 82.7 | 18.5 | 32.0 | 54.4 |
| × | × | 336 | 96.4 | 56.4 | 42.2 | 82.6 | 18.5 | 32.3 | 54.7 |
| ✓ | × | 336 | **99.0** | 75.7 | 53.4 | **86.9** | 26.2 | 29.7 | 61.8 |
| ✓ | ✓ | 336 | **99.0** | **76.7** | **58.7** | 84.3 | **27.0** | **33.2** | **63.2** |

## 6 ABLATIONS AND ANALYSIS

We perform comprehensive ablation studies to validate the effectiveness of multi-scale token learning and multi-scale representation alignment, as well as the designs of the multi-scale learnable tokens. To control the training budget, the ablation studies are conducted on a small-scale image generation dataset (with approximately 5 million image text pairs), and a small DiT structure (*i.e.,* SANA-1.6B (Xie et al., 2024a)).

### 6.1 ABLATION ON MULTI-SCALE LEANABLE TOKENS

We first conducted an ablation study on multi-scale learnable tokens. According to Table 3, Compared with Single-scale as baseline, we found that compared with Multi-scale, it improved by 8.4% on GenEval, 2.2% on Single Object and 12.8% on Counting, showing better generated image quality and prompt following ability.

### 6.2 ABLATION ON MULTI-SCALE REPRESENTATION ALIGNMENT

Based on multi-scale learnable tokens, we further conducted ablation experiments on multi-scale representation alignment. According to Table 3, we can find that representation alignment plays a role in further improving the generation effect. Multi-scale representation alignment improves the overall effect of the model by 1.5% under the GenEval measurement. A qualitative comparison is shown in Figure 5(a).

Table 4: Ablation studies on combinations of multiple scales and the number of overall tokens.

| Scales Combination | Aspect Ratio | Number of Queries | MJHQ FID)↓ | GenEval (Overall)↑ |
|---|---|---|---|---|
| 1.0x | 1:1 | 256 | 6.87 | 0.54 |
| 1.0x | any | 256 | 6.85 | 0.57 |
| 0.25x, 1.0x | any | 272 | 6.95 | 0.61 |
| 0.5x, 1.0x | any | 320 | 5.84 | 0.60 |
| 0.25x, 0.5x, 1.0x | any | 336 | **5.53** | **0.64** |
| 0.25x, 0.5x, 1.0x | any | 512 | 5.58 | **0.64** |
| 0.25x, 0.5x, 1.0x | any | 1024 | 5.69 | 0.63 |

## 6.3 ABLATION ON DESIGNS OF MULTI-SCALE LEANABLE TOKENS

We then conduct an experimental study on the combination of different scales. We start with a single scale (1.0x) and gradually add more and smaller scales (0.5x, 0.25x). And to further verify that our method is valid for more aspect ratios, we first expand the fixed 512x512 generation size to the native aspect ratio generation sizes, which is achieved by smart resizing (Bai et al., 2025) and constructing ratio buckets.

As shown in Table 4, adding a smaller resolution at the front position can significantly improve the generated quality. Adding 0.25x and 0.5x scales alone can bring 3% to 4% improvement, while adding them together can further bring a slight improvement.

The proposed multi-scale method actually increases the number of learnable tokens. To study the relationship between the number of tokens and the generation effect, we compared the experimental results with those of the single-scale method (Pan et al., 2025), as shown in Figure 5(b). We can see that in the existing single-scale learnable tokens method, the generation quality does not continue to improve effectively as the number of tokens increases (after exceeding 256). In contrast, multi-scale learnable tokens can achieve better generation results by introducing only a small number of additional tokens. The above experiments show that the multi-scale method can make tokens be used more efficiently as a bridge between understanding and generation.

## 7 DISCUSSION

**Limitations** Although the proposed method has greatly improved the quality of text-to-image generation, it is limited by the fact that the editing datasets are currently open source datasets and some datasets are not of high quality, so the editing ability may be limited in some cases. This may be further improved by increasing the number of datasets and filtering criteria.

**Conclusions** In this paper, we mainly explore the role of multi-scale query learning in unified MLLM. Based on the existing method of building unified MLLM with leanable tokens, we proposed grouped tokens to generate images progressively from low resolution to high resolution, and improved the consistency of generated multiscale images by aligning the middle layer features of the multi-scale diffusion model. After detailed experiments, we verified that this token arrangement can efficiently extract the information required for generation from MLLM. We verified the effectiveness of the method in text-to-image task, image style transfer, and fine-grained editing, while maintaining the understanding performance of the original MLLM unchanged.

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
