# OpenReview forum: "HieraQuery: Bridging Multimodal Understanding and High-Quality Generation through Multi-Scale Query Learning"
_ICLR.cc/2026/Conference — Submitted to ICLR 2026_

### Official Review · Reviewer_snFi · 2025-10-27

**Soundness:** 3
**Presentation:** 3
**Contribution:** 2
**Rating:** 4
**Confidence:** 4

**Summary:**

The paper proposes a unified autoregressive multimodal LLM framework. It introduces hierarchical learnable visual queries for coarse-to-fine image generation: preceding queries handle low-resolution global structures, while subsequent ones refine high-resolution details. A multi-scale representation alignment strategy ensures cross-scale consistency and accelerates training.

**Strengths:**

1. The work shows strong technical quality through comprehensive experiments on benchmarks like GenEval, MJHQ-FID, and GEdit-Bench, outperforming SOTA unified models. Ablations validate the multi-scale query and alignment strategies.
2. HieraQuery advances unified multimodal LLMs, and its scalable multi-scale approach addresses bottlenecks in error accumulation and resolution handling, benefiting the AI community in developing more versatile vision-language models.

**Weaknesses:**

1. Table 1 shows noticeable drops in understanding benchmarks compared to MetaQuery (which uses a similar base): MMB decreases from 83.5 to 80.7, MMMU from 58.6 to 54.3. MM-Vet improves slightly (66.6 to 74.0), but the overall trend suggests the multi-scale query integration may interfere with the LLM's text-visual alignment for understanding tasks.
2. The core contribution—multi-scale queries for coarse-to-fine generation—builds directly on MetaQuery by hierarchizing queries. While this addresses MetaQuery's plateauing with token count, it resembles existing multi-scale generation techniques, such as  *Chain-of-Sight* mentioned in the paper. This makes the novelty appear incremental rather than transformative for unified MLLMs.
3. Further detailed ablation studies are missing. For example, no quantitative analysis is provided on how performance and computational cost trade off as $K$ in $S= \\{s_1, s_2, . . . , s_K\\}$ in increases. Without this analysis, it is unclear whether increasing $K$ yields diminishing returns, introduces redundancy, or disproportionately raises inference latency.

**Questions:**

1. Could the authors elaborate on how HieraQuery's hierarchical queries differ mechanistically from multi-resolution approaches in diffusion models? A detailed comparison, perhaps with pseudo-code or diagrams, could clarify if this is a truly novel integration or an incremental extension, potentially strengthening my view on the contribution's originality.
2. In Table 1, HieraQuery shows slightly lower scores on understanding benchmarks. Since the MLLM (Ming-Lite-Omni) is frozen during training, could this be causing a degradation?
3. Minor issue: There is a redundant writing issue in L267-L269.

---

> ### Author Response · Authors · 2025-12-04
>
> **Q1: Table 1 shows noticeable drops in understanding benchmarks compared to MetaQuery.**
>
> We appreciate the reviewer’s insightful comment and we would like to clarify that, following MetaQuery’s training strategy, our approach **keeps the MLLM frozen during image generation training**. As a result, the visual **understanding capability of the MLLM remains unchanged**, and no degradation in understanding accuracy occurs due to generation optimization.
>
> The differences in understanding performance observed in Table 1 stem from the fact that MetaQuery and HieraQuery are built **upon different base MLLMs**, rather than from any trade-off introduced by our method. Both methods preserve the original understanding capabilities of their respective frozen MLLMs.
>
> To ensure a fair comparison focused specifically on generation improvements, **we align the backbone MLLM in our ablation studies**. Under this controlled setting, the enhanced generation quality is achieved without compromising understanding performance—demonstrating that our method improves generation strictly through architectural or training-level innovations, not at the cost of understanding.
>
> **Q2: The core contribution—multi-scale queries for coarse-to-fine generation—builds directly on MetaQuery by hierarchizing queries. While this addresses MetaQuery's plateauing with token count, it resembles existing multi-scale generation techniques, such as Chain-of-Sight mentioned in the paper. This makes the novelty appear incremental rather than transformative for unified MLLMs.**
>
> Thank you for the feedback. We agree that several methods explore hierarchical visual modeling, and we appreciate the opportunity to clarify the conceptual and architecture distinctions of HieraQuery.
>
> While VAR adopts a **sequential generation paradigm**, where each scale is autoregressively produced based on the previous ones, HieraQuery enables single-step, parallel multi-scale generation. During training, multi-scale supervision is imposed by grouping learnable queries to produce different resolutions in parallel. At inference time, all query groups are fed simultaneously, and the finest scale tokens can directly generate high-resolution content by attending to coarse-level semantic information encoded within the query states. This enables end-to-end, **single-step multi-scale generation with strong cross-scale coherence**, distinguishing our approach from VAR's dependency chain and highlighting its architectural novelty in unifying multi-scale understanding within a unified query space.
>
> Moreover, Chain-of-Sight is proposed mainly for multi-scale visual understanding, which introduces multi-scale learnable queries and cross-attentions to extract visual cues from input images. In contrast, HieraQuery targets generative modeling, where our hierarchical query structure is explicitly optimized for coherent multi-scale image synthesis, not reasoning or perception.
>
> Thus, despite superficial similarities in using hierarchical queries, HieraQuery introduces a distinct paradigm—one that rethinks how unified MLLMs can leverage scale hierarchy through structured concurrency rather than sequential processing.
>
> We will include a conceptual comparison table in the final version to further clarify these differences.
>
> **Q3: Further detailed ablation studies are missing. For example, increasing more scales.**
>
> We acknowledge the suggestion to explore additional scales; however, as analyzed in response to 5WH4-Q3, training a 3-scale token model already incurs 1.63× the computational cost of the baseline. **Further increasing the number of scales would impose a substantially higher training burden, leading to diminishing returns in generation quality relative to the escalating computational demands**. Given the need for practical scalability and efficiency, we prioritize a balanced trade-off between performance gains and computational feasibility, thus limiting the current design to three scales.
>
> **Q4: Could the authors elaborate on how HieraQuery's hierarchical queries differ mechanistically from multi-resolution approaches in diffusion models?**
>
> Unlike diffusion models that process multi-resolution features through U-Net skip connections or progressive denoising, HieraQuery introduces structured learnable queries grouped by scale. These hierarchical queries are trained to produce images at different resolutions in parallel and enforce cross-scale semantic coherence via attention. Crucially, at inference, all query groups are processed in a single step—without iterative denoising or feature chaining. This enables end-to-end high-resolution generation, where fine-scale tokens attend directly to coarse semantics encoded in the query space, rather than relying on spatial feature maps from earlier denoising steps.
>
> **Q5: Minor issue: There is a redundant writing issue in L267-L269.**
>
> Thank you for pointing that out. We have carefully reviewed the text again and corrected the typos.

---

### Official Review · Reviewer_5WH4 · 2025-10-30

**Soundness:** 3
**Presentation:** 3
**Contribution:** 3
**Rating:** 4
**Confidence:** 4

**Summary:**

The paper HieraQuery introduces a hierarchical query modeling framework to improve unified multimodal generation.
Instead of relying on a single query token set, it designs multi-scale queries—from coarse to fine—to progressively build images, where coarse queries capture global semantics and fine queries refine local details.
It further aligns features across scales through multi-scale representation alignment, ensuring semantic consistency between the diffusion backbone and the visual encoder.
Built on a frozen MLLM (Ming-Lite-Omni) with diffusion backbones like SD3-Medium and SANA-1.6B, the system achieves better image fidelity and structure, improving FID, GenEval, and DPG performance while maintaining strong multimodal understanding.

**Strengths:**

1.The hierarchical query framework provides a clear and modular architecture that systematically connects multimodal understanding with coarse-to-fine generation.
2.The multi-scale query learning and cross-scale alignment improve the balance between global semantic coherence and fine-grained visual fidelity.
3.The model demonstrates strong results on both text-to-image and editing tasks, showing good generality across benchmarks.
4.The paper presents extensive experiments as well as thorough ablation studies.

**Weaknesses:**

1. There is a typo in Figure 3: it should be qualitative and not quantitative.
2. The qualitative examples are too simple. For compositional task, please include more complex prompts.
3. The paper does not provide a quantitative analysis of additional computational cost, such as FLOPs, memory, or end-to-end inference time, so the efficiency trade-offs of the hierarchical design remain unclear.
4. The paper mainly reports benchmark scores (FID, GenEval, DPG) without perceptual or human evaluation on image coherence, consistency, or failure modes.
5. The understanding–generation balance is claimed but not verified with detailed breakdowns; it remains unclear whether improvements in generation come at the cost of understanding accuracy.

**Questions:**

In the ablation study, several configurations yield very similar quantitative results. How did the authors determine which setting is optimal in the first place? Was the choice based on efficiency (e.g., computational cost, FLOPs, or latency), stability, or qualitative evaluation? Providing a clearer selection criterion would help assess the robustness of the claimed improvements.

---

> ### Author Response · Authors · 2025-12-04
>
> **Q1: typo.**
>
> Thank you for pointing that out. We have carefully reviewed the text again and corrected the typos.
>
> **Q2: For compositional task, please include more complex prompts.**
>
> Thank you for the suggestion. In the new version, we will expand the evaluation to include substantially more complex prompts and report detailed results.
>
> **Q3: A quantitative analysis of additional computational cost.**
>
> We thank the reviewer for the suggestion to report training cost and inference speed. We have computed the FLOPs of both HieraQueries (multi-scale token) and the single-scale token baseline.
>
> **Due to space limitations, I refer you to our response to Reviewer 4tUX-Q4 for further details.**
>
> In summary, the additional inference and training costs are very limited, with the multi-scale approach incurring **only a ~3% increase in inference** FLOPs and 1.63x cost in training, demonstrating its efficiency and practicality.
>
> **Q4: Lack of human evaluation on image coherence, consistency, or failure modes.**
>
> _Human Evaluation_
>
> As a supplementary evaluation to provide a more intuitive assessment of HieraQuery's performance, we conducted a user study. Given that human evaluation is time-consuming and resource-intensive, we selected the top-performing variants of MetaQuery for comparison—specifically, our trained baseline model and the open-source UniPic2-MetaQuery-9B (a non-official reimplementation, as no official checkpoint has been released).
>
> For each model, we generated images using a consistent set of 300 prompts drawn from established benchmarks: PartiPrompt (Yu et al., 2022), MJHQ-30K (Li et al., 2024a), and GenEval (Ghosh et al., 2024). The resulting images were then presented to human evaluators in a blinded study. Participants were instructed to rank the models based on two criteria: (1) the perceptual quality of the generated images (e.g., clarity, realism, and visual coherence), and (2) the alignment accuracy between the input text prompts and the generated content.
>
> | Baseline Model | Quality (HieraQuery vs. Baseline) | Alignment (HieraQuery vs. Baseline) |
> | -------- | -------- | -------- |
> | MetaQuery | 54.8% vs 45.2%  | 68.4% vs 31.6% |
> | UniPic2-MetaQuery-9B | 53.7% vs 46.3%  | 68.0% vs 32.0% |
>
> HieraQuery consistently outperforms both baselines. While it maintains a steady advantage in Image Quality, it demonstrates superior performance in Prompt-Image Alignment, being preferred in approximately 68% of cases against both comparison models.
>
> _Failure Modes_
>
> The primary failure modes are observed in scenarios requiring high fidelity, specifically involving character appendages (e.g., hands/fingers) and intricate elements like complex textual arrangement, and will include in the camera ready version.
>
> **Q5: it remains unclear whether improvements in generation come at the cost of understanding accuracy.**
>
> We appreciate the reviewer’s insightful comment regarding the understanding–generation balance. We would like to clarify that, following MetaQuery’s training strategy, our approach **keeps the MLLM frozen during image generation training**. As a result, the visual **understanding capability of the MLLM remains unchanged**, and no degradation in understanding accuracy occurs due to generation optimization.
>
> The differences in understanding performance observed in Table 1 stem from the fact that MetaQuery and HieraQuery are built **upon different base MLLMs**, rather than from any trade-off introduced by our method. Both methods preserve the original understanding capabilities of their respective frozen MLLMs.
>
> To ensure a fair comparison focused specifically on generation improvements, **we align the backbone MLLM in our ablation studies**. Under this controlled setting, the enhanced generation quality is achieved without compromising understanding performance—demonstrating that our method improves generation strictly through architectural or training-level innovations, not at the cost of understanding.
>
>
> **Q6: In the ablation study, several configurations yield very similar quantitative results. How did the authors determine which setting is optimal in the first place?**
>
> You are correct; we prioritized computational efficiency when quantitative results were comparable. While Table 3 validates the effectiveness of our core modules, Table 4 reveals that increasing the token count in Multi-scale Query Learning saturates performance while significantly raising training costs (detailed in Response to 4tUX-Q4). Therefore, following Occam's Razor, we selected the configuration with fewer tokens to minimize computational overhead without sacrificing accuracy.

---

### Official Review · Reviewer_4tUX · 2025-10-30

**Soundness:** 3
**Presentation:** 2
**Contribution:** 2
**Rating:** 4
**Confidence:** 2

**Summary:**

This paper propose HieraQuery, a multi-scale query learning method for high-quality visual generation, which leverages a hierarchy of learnable visual queries for generation in a coarse-to-fine manner and includes a multi-scale representation alignment strategy for cross-scale consistency and convergence acceleration. Extensive experiments is conducted and the proposed method demonstrate the performance improvement on the visual generation capability.

**Strengths:**

1. Query learning is worth studying and have certain commonality for multimodal tasks, which is brave and novel.
2. The experiment result is solid under text-to-image generation, image style transfer and fine-grained editing.
3. Convincing visualization examples are provided to prove the effectiveness of the proposed method.

**Weaknesses:**

1. Lack of concept comparison framework between the baseline methods and proposed HieraQuery method. Providing this may help better presentation.
2. Comparison on understanding benchmarks should includes used MLLM backbone.
3. Several writing typos: “benchmarsk” at line 265 and MJHQ FID) at Table 4.

**Questions:**

1. See weakness.
2. The training cost and inference speed between proposed method and baselines may require to provide.
Overall, I think this work is novel, but with several weaknesses on writing and experiment. So I tend to adjust my score and confidence based on the author's response and the opinions of other reviewers.

---

> ### Author Response · Authors · 2025-12-04
>
> **Q1: Lack of concept comparison framework.**
>
> Thank you for your suggestion. We agree that adding a conceptual comparison framework between the baseline methods and HieraQuery will improve the presentation, and we will include it in the final version.
>
> Thank you for your suggestion. We agree that a clear conceptual comparison enhances clarity, and we will include such a framework in the final version.
>
> Conceptually, HieraQuery differs from existing methods in the following key aspects:
>
> **Compared to the unified baseline (e.g., MetaQuery)**
>
> While MetaQuery uses a fixed set of learnable queries and show diminishing returns as token count increases, HieraQuery introduces a multi-scale query grouping mechanism that enables effective scaling with more tokens. This breaks the performance bottleneck observed in standard unified architectures, where additional capacity does not translate into better generation quality.
>
> **Compared to multi-scale generation methods like VAR**
>
> VAR follows a sequential, autoregressive paradigm, generating each scale conditioned on the previous one. In contrast, HieraQuery adopts a single-step, parallel multi-scale generation scheme, all scale-specific query groups are processed simultaneously, and the finest-scale tokens can directly generate high-resolution content by attending to coarse-level semantics.
>
> **Q2: Comparison on understanding benchmarks should includes used MLLM backbone.**
>
> Thank you for the feedback. HieraQuery does not alter the underlying comprehension capability of the model, and the understanding results in Table 1 reflect the used MLLM backbone. We will clarify this point more explicitly in the next version.
>
> **Q3: Several writing typos.**
>
> Thank you for pointing that out. We have carefully reviewed the text again and corrected the typos.
>
> **Q4: The training cost and inference speed between proposed method and baselines may require to provide.**
>
> We thank the reviewer for the suggestion to report training cost and inference speed. We have computed the FLOPs of both HieraQueries (multi-scale token) and the single-scale token baseline.
>
> The inference cost consists of two parts: the LLM (including additional MLPs and connectors) and the diffusion module.
> The computational time of the LLM scales with the number of tokens.
> For the diffusion component, **although multi-scale training** involves feeding multiple scales sequentially into the DiT, during **inference only the largest scale** is used as input to the DiT.
>
> _Inference_
>
> For the LLM part, the FLOPs are 3.57 T (multi-scale) vs 2.81 T (single-scale). For the DiT part, each scale requires 0.75 T FLOPs per step and 30 steps for inference, resulting in 22.5 T FLOPs per scale. This gives total FLOPs 0.75T * 30 + 3.57 T= 26.07 T (multi-scale) vs 25.31 T (single-scale), i.e., m**ulti-scale incurs only ~3% higher computational cost at inference.**
>
> |  | single-scale (TFLOPs) | multi-scale (TFLOPs) | extra cost introduced by multiscale (TFLOPs)
> | -------- | -------- | -------- | -------- |
> | LLM | 2.81  | 3.57 | 0.76 |
> | Diffusion | 22.5  | 22.5 | 0 |
> | Overall | 25.31  | 26.07 | 0.76 (~3%) |
>
> _Training_
>
> For the DiT module, multi-scale token training requires 3 forward passes compared to 1 for single-scale. LLM is still computed once in both cases. This results in forward FLOPs of 3.57 T + 0.75 T * 3 = 5.82 T (multi-scale) vs 2.81 T + 0.75T = 3.56 T (single-scale), the training cost becomes 1.63 times the original.
>
> |  | single-scale (TFLOPs) | multi-scale (TFLOPs) | extra cost introduced by multiscale (TFLOPs)
> | -------- | -------- | -------- | -------- |
> | LLM | 2.81  | 3.57 | 0.76 |
> | Diffusion | 0.75  | 2.25 | 1.5 |
> | Overall | 3.56  | 5.82 | 2.26 (1.63x) |
>
>
> In summary, the additional inference and training costs are very limited, with the multi-scale approach incurring **only a ~3% increase in inference** FLOPs and 1.63x cost in training, demonstrating its efficiency and practicality.

---

### Official Review · Reviewer_cq6g · 2025-11-03

**Soundness:** 2
**Presentation:** 2
**Contribution:** 2
**Rating:** 4
**Confidence:** 5

**Summary:**

This paper proposed HieraQuery, which extends the method of MetaQuery to a multiscale paradigm by generating images of multiple resolutions on different sets of conditioning queries. To enforce cross-scale consistency, REPA is applied to different levels, so the DiT features from all these scales are aligned to the same semantic representation. Beyond image generation, the framework is also adapted to image editing, with a progressive reconstruction-and-then-editing paradigm. In the experiment, performance gain by using the multiscale queries is observed on GenEval.

**Strengths:**

1. The motivation is clear, and the paper is easy to follow.

2. The proposed framework, HieraQuery,  comprehensively supports image understaning, generation, and editing.

3. HieraQuery achieves competitive performances on all three tasks studied.

**Weaknesses:**

1. Technical contribution is limited, since the ideas of multiscale generation and representation alignment are already proposed in existing works, i.e, VAR and REPA.

2. Although this paper studies unified models that cover understanding, generation, and editing. The advantage of multiscale queries is only experimentally supported by results on the image generation benchmark---GenEval.

3. Some results are expected but missed in the ablation study:

    **a.** In Table 3, the result of applying REPA to the single-scale setting is expected.
    **b.** DiT is shared across different scales. Will some scale-specific parameters benefit the final performance?

4. In the 2nd and 3rd training stages, only image-to-image losses are applied, which seems to damage the text-to-image generation capability of the model.

**Questions:**

Some details are missing:
1. SD3-Medium and SANA-1.6B are both studied. But it is unclear which model is used in Table 1&2.
2.  Can the authors provide more details on editing? For example, are the same model parameters used for both generation and editing? Are VAE latents fed to the DiT to ensure consistency between source and edited images?

---

> ### Author Response · Authors · 2025-12-04
>
> **Q1: Limited novelty—core ideas already present in VAR and REPA.**
>
> We thank the reviewer for pointing out the relation of HieraQuery to VAR and REPA. While these works inspired our research, HieraQuery presents a disctinct motivation and solution in the following aspects.
>
> **Single-step multi-scale generation in HieraQuery vs. Sequential next-scale prediction in VAR**
>
> While both VAR and HieraQuery involve hierarchical visual modeling, they differ fundamentally in methodology and objective. VAR adopts a **sequential generation paradigm**, where each scale is autoregressively produced based on the previous ones.
>
> In contrast, HieraQuery leverages multi-scale supervision during training by grouping learnable queries to produce different resolutions in parallel. Crucially, at inference time, all query groups are fed simultaneously, and the finest scale tokens can directly generate high-resolution content by attending to coarse-level semantic information encoded within the query states. This enables end-to-end, **single-step multi-scale generation with strong cross-scale coherence**, distinguishing our approach from VAR's dependency chain and highlighting its architectural novelty in unifying multi-scale understanding within a unified query space.
>
> **Multi-scale consistency vs. REPA**
>
> REPA aims to align intermediate features of diffusion models with frozen DINOv2 features for faster convergence. In contrast, our multi-scale consistency loss serves a different purpose. It enfoces intra-model semantic coherence by explicitly aligning the representations of learnable query groups across scales, ensuring that the coarse-scale semantics can be effectively propagated to finer ones during joint generation.
> As shown in the Table below, REPA improves GenEval score by 0.8pt (55.2 vs 54.4) on single scale scenario, while multi-scale consistency loss improves GenEval score by 1.4pt (63.2 vs 61.8), showing more effectiveness under the setting of multi-scale generation.
>
> **Q2: The benefit of multiscale queries in unified models is evidenced only by GenEval image generation results.**
>
> Although the paper emphasizes the advantage of multiscale queries mainly through the GenEval image generation benchmark, Table 1 demonstrates that our approach achieves a clear and consistent improvement in **FID**—reducing it from **6.02 to 4.85**, a **19%** decrease—while also delivering competitive performance on **DPG-bench**, where our method improves the score from **82.0 to 84.2** compared to MetaQuery, reflecting a notable enhancement in generation quality and evaluation robustness.
>
> **Q3: REPA single-scale results missing; would scale-specific DiT parameters help?**
>
> _REPA single-scale results_
>
> Applying Multi-Scale Representation Alignment to the 256‑token single-scale setting reduces it to REPA; we new add its experimental results in Table 3 as follows.
>
> | No. | Multi-scale Query Learning (M.)   Multi-Scale Representation Alignment (A.) | Num tokens | Single Obj. | Two Obj. | Counting | Colors | Position | Color Attri.| Overal |
> | -------- | -------- | -------- | -------- | -------- | -------- | -------- | -------- | -------- | -------- |
> | 1 | w/o w/o  | 256 | 96.8 | 55.8 | 40.6 | 82.7 | 18.5 | 32.0 | 54.4 |
> | 2 (**new**, single-sclae + REPA) | w/o w/  | 256 | 96.9 | 56.3 | 41.5 | 82.3 | 20.1 | 34.0 | 55.2 |
> | 3 | w/o w/o  | _336_ | _96.4_ | _56.4_ | _42.2_ | _82.6_ | _18.5_ | _32.3_ | _54.7_ |
> | 4 | w/  w/o  | 336 (16 + 64 + 256) | **99.0** | 75.7 | 53.4 | **86.9** | 26.2 | 29.7 | 61.8 |
> |5 | w/  w/   | 336 (16 + 64 + 256) | **99.0** | **76.7** | **58.7** | 84.3 | **27.0** | **33.2** | **63.2** |
>
> The single-scale + REPA approach shows a slight improvement over single-scale without REPA (54.4 → 55.2).
>
> _Would scale-specific DiT parameters help?_
>
> While scale‑specific parameters might potentially improve performance, they would also increase training and inference overhead, which may be less efficient than simply scaling up DiT directly. In contrast, our method incurs only minimal overhead for both training and inference.
>
> **Q4: In the 2nd and 3rd training stages, only image-to-image losses are applied, which seems to damage the text-to-image generation capability of the model.**
>
> While the new data is emphasized in the 2nd and 3rd training stages, we still retain a portion of the original text-to-image (T2I) data to preserve the model's basic text-to-image generation capability. We will clarify this in the camera-ready version.
>
> **Q5: SD3-Medium and SANA-1.6B are both studied. But it is unclear which model is used in Table 1&2.**
>
> As shown in Line 237, in Table 1 we used the SD3-Medium model. For the ablation study, we adopted SANA-1.6B (see Line 415) in order to enable a fair comparison with existing methods. We will make this distinction more explicit and clear in the final version of the paper to avoid any misunderstanding.

---

### Meta-Review · Area_Chair_ceno · 2026-01-08

**Summary:**

This work proposes multi-score query learning (HieraQuery) to bridge multimodal understanding and generation in unified models. Four reviewers all gave the initial scores of borderline reject, with various concerns. A common concern is the incremental novelty and limited technical contribution of the proposed approach. Other concerns include the insufficient experimental validation, understanding-generation trade-off, etc.

**Reviewer Concerns:**

Some reviewer concerns like the understanding-generation tradeoff and some required ablation experiments or experimental results are addressed during the rebuttal. However, it seems that the major concern of limited novelty and technical contribution is not fully addressed with convincing evidence.

**Reviewer Scores:**

Given the unsolved issue on limited technical contribution and novelty, it is likely that most reviewers will keep the original score of borderline reject after rebuttal.

---

### Decision · Program_Chairs · 2026-01-26

Reject